# Evaluation of Turbulent Flux Parameterizations over a

# **2 Continental Glacier on the Tibetan Plateau**

- 3 Yichan Xu<sup>1,2</sup>, Meilin Zhu<sup>2,\*</sup>, Jing Gao<sup>3,\*</sup>, Daijiao Lin<sup>1,2</sup>, Lei Hui<sup>4</sup>, Fei Zhu<sup>5</sup>, Huabiao
- 4 Zhao<sup>3</sup>, Fengying Zhang<sup>1,2</sup>
- <sup>1</sup>College of Earth and Environmental Sciences, Lanzhou University, Lanzhou, 730000, China
- <sup>2</sup>Center for the Pan-Third Pole Environment, Lanzhou University, Lanzhou, 730000, China
- 3State Key Laboratory of Tibetan Plateau Earth System, Environment, and Resources (TPESER),
- Institute of Tibetan Plateau Research, Chinese Academy of Sciences, Beijing 100101, China
- <sup>4</sup>Danghe River Basin Water Resources Administration Bureau, Dunhuang, Gansu, 736200, China
- <sup>5</sup>College of Surveying and Geo-Informatics, Tongji University, Shanghai, 200092, China
- Correspondence to: Meilin Zhu (zhuml@lzu.edu.cn) and Jing Gao (gaojing@itpcas.ac.cn)

Abstract. A lack of observations of turbulent fluxes over continental glaciers limits our 13 understanding of the mechanisms that control glacier variations and associated water resource 14 changes across the Tibetan Plateau (TP). Here, we present the first comprehensive analysis of turbulent 15 flux characteristics and a systematic evaluation of turbulent flux methods for a continental glacier on the 16 TP, using eddy covariance observations from the Dunde Glacier (May-October, 2023). The Dunde 17 Glacier persistently lost energy through latent heat flux (mean LE: -10.34 W m<sup>-2</sup>) and gained energy via 18 sensible heat flux (mean H: 6.93 W m<sup>-2</sup>), with pronounced seasonal and diurnal variability. On the basis 19 of these measured data, we tested five turbulent flux methods for the Dunde Glacier, including those 20 derived from katabatic flow models, simplified Monin-Obukhov similarity theory without stability 21 corrections, Monin-Obukhov similarity theory with stability corrections using two different bulk 22 Richardson numbers, and the Monin-Obukhov similarity theory with universal stability functions. 23 Among all schemes, the Monin-Obukhov similarity theory with universal stability functions achieved 24 the highest accuracy for both H and LE at different timescales. We further evaluated the performance of 25 these parameterizations in energy and mass balance modeling. Our results show that the recalibrated 26 turbulent flux parameterizations are an effective approach for improving the accuracy of modelled glacier 27 energy and mass balance, and that the Monin-Obukhov similarity theory with universal stability 28 functions yielded the best simulation performance for modelled glacier mass balance. We also found that 29 the Dunde Glacier experienced a sharp increase in H and reversal in LE during a humid heatwave event, 30 shifting from a negative total turbulent flux under the mean climate condition to positive values during

https://doi.org/10.5194/egusphere-2025-4227 Preprint. Discussion started: 15 October 2025 © Author(s) 2025. CC BY 4.0 License.

the extreme event. However, none of the turbulent flux methods fully captured the high values that occurred during the extreme weather and climate event, indicating that there is currently an underestimation of the contribution of turbulent fluxes to glacier melt energy. These findings advance our knowledge of turbulent fluxes for continental glaciers on the TP and provide important guidance for the improvement of glacier models.

#### 1 Introduction

49

5758

Turbulent fluxes, including latent heat flux (LE) and sensible heat flux (H), are critical components of the surface energy balance (SEB) of glaciers and markedly influence glacier mass-loss processes. Research on the Tibetan Plateau (TP) shows that turbulent fluxes contribute approximately 14%-35% of the glacial energy budget (Li et al., 2011; Mandal et al., 2022; Zhang et al., 2013; Zhu et al., 2017), serving as the primary energy source sustaining sublimation-driven mass loss, particularly during the cold season (October-May) (Azam et al., 2014a; Potocki et al., 2022). In addition, turbulent fluxes are one of the primary energy sources driving extreme glacier loss events (Thibert et al., 2018). Such events enhance the provision of water resources to downstream regions (Gao et al., 2019; Yao et al., 2022), altering hydrological processes therein (Radić and Hock, 2011; Zhu et al., 2024a); they may also be linked to glacier-related disasters, such as glacier avalanches (Zhao et al., 2022). Under ongoing global warming, the frequencies and intensities of extreme weather and climate events are projected to increase, likely contributing to severe future glacier mass loss owing to enhanced turbulent fluxes (Brun et al., 2018; Duan et al., 2012; Hugonnet et al., 2021; Yao et al., 2012). Therefore, a comprehensive understanding of the magnitude and variability of turbulent fluxes is essential to accurately quantify glacier SEB and project future glacier mass-balance changes. Currently, glacier turbulent flux data are obtained through two main approaches: direct measurement using the eddy covariance (EC) method and estimation using numerical models. Turbulent flux observations over glaciers on the TP remain scarce, primarily owing to the difficulty of obtaining sustained measurements on glacier surfaces (Box and Steffen, 2001; Cullen et al., 2007; Kuipers Munneke et al., 2009). Notably, Yang et al. (2011) carried out observations over a 3-month period on the Parlung No. 4 Glacier, a maritime glacier on the southeastern TP characterized by high air temperatures and precipitation. They reported that during summer, the observed average H reached 28 W m<sup>-2</sup>, whereas LE remained relatively low at -1 W m<sup>-2</sup>. However, to date, there have been no reported observations of turbulent fluxes over continental glaciers on the TP characterized by low temperatures and precipitation, one of the most widely distributed glacier types on the TP. This data gap has resulted in an inadequate understanding of the energy balance processes of TP glaciers, which, in turn, is important to understand the climatic mechanisms controlling glacier variations across the TP.

https://doi.org/10.5194/egusphere-2025-4227 Preprint. Discussion started: 15 October 2025 © Author(s) 2025. CC BY 4.0 License.

Owing to the scarcity of observational data, numerical modeling has become the primary method by which glacier turbulent fluxes are estimated on the TP (Mölg et al., 2012; Yang et al., 2011; Zhu et al., 2018). Current turbulent flux modeling approaches predominantly rely on Monin-Obukhov similarity theory (MOST) (Monin and Obukhov, 1954), which provides accurate simulations of turbulent fluxes but is computationally intensive. In addition to sophisticated turbulent flux models that strictly implement MOST (Hock and Holmgren, 2005; Yang et al., 2002), various simplified turbulent flux parameterizations have been developed to derive LE and H. To improve computational efficiency, Oerlemans (2000) developed a highly simplified derivative model based on MOST, eliminating typically required iterative calculations. Essery and Etchevers (2004) and Suter et al. (2004) introduced stability functions to simulate turbulent fluxes. Oerlemans and Grisogono (2002) developed a turbulent flux model adapted to local glacier climatic conditions, which considers the influence of glacier wind on turbulent fluxes. However, the modeled turbulent fluxes for a given glacier tend to differ substantially across different turbulent flux parameterizations (Radić et al., 2017). Taking the Zhadang Glacier as an example, Zhang et al. (2016) found that the turbulent heat fluxes were 13.4 and 5.7 W m<sup>-2</sup> for the winter and summer seasons, respectively, during the period 2011-2014; meanwhile, Li et al. (2014) reported that the turbulent heat fluxes were -1 and -11 W m<sup>-2</sup> for the winter and summer seasons, respectively, during the period 2006-2011. The important reasons of such difference are the lack of observational data which can be used to calibrate the turbulent flux parameterizations. evaluations of the accuracy of those different turbulent flux parameterizations remain limited on the TP. Guo et al. (2011) evaluated three turbulent flux parameterizations for the Parlung No.4 Glacier (maritime glacier) and found that the scheme proposed by Yang et al. (2002) produced lower errors in turbulent flux estimates during both individual melt phases and the entire ablation season. Compared with the other two schemes evaluated (Andreas, 1987; Smeets and van den Broeke, 2008), the mean absolute deviation (MAD) was reduced by approximately 12-29%. Yet, no comprehensive analysis exists for continental glaciers and no evaluation of the impact of the various parameterizations on glacier mass balance simulation on the TP. And we still do not know if these parameterizations can accurately capture the influence of extreme weather and climate events on turbulent fluxes. These uncertainties hinder accurate simulations of glacier energy and mass balances and impede a comprehensive understanding of accumulation and ablation processes and the physical relationship between glaciers and climate (Zhu et al., 2023).

Here we provide the first systematically analyze of meteorological and glacier mass balance observations and direct eddy-covariance-based turbulent flux measurements at the Dunde Glacier in the Qilian Mountains, the northeastern TP, to evaluate the applicability of multiple widely used turbulent flux methods over continental glaciers on the TP. We systematically analyze observed and modeled turbulent fluxes across multiple temporal scales, including seasonal, daily mean, diurnal, and notably under extreme weather conditions, to test the model robustness. Finally, we assess the influence of the various turbulent flux parameterizations on the simulated glacier mass balance by implementing the parameterizations in the energy and mass balance model. The findings allow us to improve the accuracy of glacier surface energy and mass balance modeling and to advance our understanding of ablation processes on continental glaciers on the TP.

# 2 Study Area and Data

The Dunde Glacier is located in the western part of the Qilian Mountains, on the northeastern TP. It is a typical continental glacier, with an elevation ranging from 4400 to 5500 m (Guo et al., 2015). The ice cap covers an area of approximately 60 km<sup>2</sup>, with an average annual elevation change rate of +0.04  $\pm$  0.15 m yr<sup>-1</sup> (Hugonnet et al., 2021; Shean et al., 2020). The region experiences pronounced annual and diurnal temperature variations, with a mean annual temperature of approximately -5 °C and annual precipitation ranging from 300 to 400 mm. Meltwater from the glacier primarily feeds into the Tataleng and Haerteng rivers, which provide an important water source to their respective drainage basins. Turbulent fluxes were measured using the eddy covariance (EC) instrument (Campbell-IRGASON), operating from 14 May until 12 October, 2023; this is a widely adopted technique in micrometeorological research that enables real-time, high-precision, and continuous monitoring of atmospheric turbulence. All turbulence raw data were collected at 10 Hz, including the three components of wind velocity, virtual temperature, and water vapor concentration. The observation site was situated on the surface of the Dunde Glacier (38°06'23" N, 96°24'54" E) at an elevation of 5317 m a.s.l., as shown in Fig. 1. In addition, air temperature (Ta), air pressure, relative humidity, wind speed and direction, incoming and outgoing shortwave radiation, incoming and outgoing longwave radiation, and glacier surface height were also measured at this site, using Campbell-ClimaVUE50 automatic weather station (AWS). Precipitation was

measured using a Geonor T-200B rain gauge at 4970 m a.s.l., and missing values in the precipitation

record were supplemented using ERA5 reanalysis data. Details of the sensors used to measure each variable are provided in Table 1.

123

125

126

127

Figure 1: (a) Locations of the Dunde Glacier (blue hexagon) and selected reference glaciers (purple triangles) on the Tibetan Plateau. (b) Topographic map of the Dunde Glacier (40 m contour intervals) and locations of AWS1 and AWS2 (red and black stars, respectively). (c-d) Photographs of AWS2, equipped with an eddy covariance (EC) system (Photo credit: Meilin Zhu).

Table 2: Characteristics of the sensors installed in the eddy covariance system to measure turbulent fluxes and meteorological variables in this study.

| Variable                                  | Symbol<br>(unit)                 | Sensor          | Accuracy                          | Range                         | Height |
|-------------------------------------------|----------------------------------|-----------------|-----------------------------------|-------------------------------|--------|
| Air temperature                           | T <sub>a</sub> (°C)              | Vaisala HMP155A | $\pm (0.055 \pm 0.0057 * T_a)$ °C | −80 to 60 °C                  | 2.17 m |
| Relative humidity                         | RH (%)                           | Vaisala HMP155A | ± 1%                              | 0%-100%                       | 2.17 m |
| Wind speed                                | $u (m s^{-1})$                   | LICOR LI-7500DS | ± 1.5%                            | $0\!-\!100~{\rm m~s^{-1}}$    | 2.05 m |
| Wind direction                            | WD (°)                           | LICOR LI-7500DS | ± 2°                              | 360 °                         | 2.05 m |
| Air pressure                              | P (pa)                           | Vaisala PTB210  | $\pm 0.5 \text{ hPa}$             | 50–1100 hPa                   | 2.05 m |
| Incoming and outgoing longwave radiation  | LWI, LWO<br>(W m <sup>-2</sup> ) | Campbell CNR4   | ± 1%                              | -250 to 250 W m <sup>-2</sup> | 1.6 m  |
| Incoming and outgoing shortwave radiation | SWI, SWO<br>(W m <sup>-2</sup> ) | Campbell CNR4   | ± 1%                              | $0-2000~{\rm W}~{\rm m}^{-2}$ | 1.6 m  |
| Precipitation                             | (mm)                             | Geonor T-200B   | $\pm~0.1\%~FS$                    | 0–600 mm                      | 1.7 m  |

## 3 Methods

The naming of the turbulent flux calculation methods in this study follows the convention of Radić et al. (2017). However, differences exist in the implementation of the  $C_{M-O}$  method (3.5). Specifically, although both our  $C_{M-O}$  method and that of Radić et al. (2017) are based on the bulk aerodynamic method combined with Monin–Obukhov similarity theory and iterative closure of the H, the source of roughness lengths is treated differently. Radić et al. (2017) prescribed fixed roughness lengths obtained as the logarithmic means of 30-min OPEC inversions under near-neutral conditions, whereas in this study, the roughness lengths are computed dynamically following Andreas (1987), as functions of the friction velocity and the Reynolds number. This modification allows the scalar roughness lengths ( $z_{0t}$  and  $z_{0e}$ ) to respond to changes in atmospheric stability and flow regime, thereby providing a more physically responsive formulation.

# **3.1.** C<sub>kat</sub> method

- The C<sub>kat</sub> method incorporates the influence of glacier winds (katabatic flows) on surface energy
- fluxes using the bulk aerodynamic method, which relates turbulent fluxes to the structure of the ambient
- atmosphere (Oerlemans and Grisogono, 2002). In a stably stratified atmosphere, it identifies H and
- vertical advection as the primary contributors to the surface energy budget; LE and H are expressed as:

$$LE = 0.622\rho L_{s/f}C_{kat}(e_a - e_s),$$
 (1)

$$H = 0.622C_pC_{kat}(T_a - T_s),$$
 (2)

- where,  $\rho$  and  $C_P$  are the density (kg·m<sup>-3</sup>) and heat capacity of air, respectively;  $L_{s/f}$  is the latent heat
- of sublimation/fusion, selected based on surface temperature;  $e_a$  and  $e_s$  are the atmospheric vapor
- pressure and saturated vapor pressure at the glacier surface (Pa), respectively;  $T_a$  and  $T_s$  denote the air
- temperature and glacier surface temperature, respectively; and  $C_{kat}$  is the katabatic bulk exchange
- coefficient, calculated using the following equation:

$$C_{kat} = -C_{tub}C_{tub_2}^2C\left(\frac{g}{T_0\gamma P_r}\right)^{1/2},$$
 (3)

- Here,  $C_{tub}$  and  $C_{tub2}$  are dimensionless empirical constants;  $T_0 = 273.15$  K;  $\gamma$  is the potential
- temperature gradient;  $P_r$  is the Prandtl number (~ 0.71); and g is gravitational acceleration (~ 9.8
- m·s<sup>-2</sup>).
- 3.2. Clog method
- This method represents a highly simplified derivative of MOST (Oerlemans, 2000). It does not
- dynamically adjust atmospheric stability, and instead retains only a linear relationship for surface fluxes;
- LE and H are expressed as:

$$LE = 0.622\rho L_{s/f}C_h u(e_a - e_s)/P,$$
 (4)

$$164 H = \rho C_p C_h u (T_a - T_s), (5)$$

- where, u is wind speed (m·s<sup>-1</sup>), P is atmospheric pressure (Pa), and  $C_h$  is the turbulent exchange
- coefficient.
- 3.3. C<sub>Rib1</sub> method

- The  $C_{Rib1}$  method replaces the Monin-Obukhov length (L) with the bulk Richardson number ( $Ri_b$ )
- to diagnose atmospheric stability (Suter et al., 2004). It replaces the stability functions (y) in MOST with
- a composite function  $(f_h(Ri_b))$ , derived from stability functions for momentum  $(\emptyset_m)$ , heat  $(\emptyset_h)$ , and
- water vapor ( $\emptyset_q$ ) (Dyer, 1974; Holtslag and Bruin, 1988). This approach avoids complex iterative
- calculations while preserving the core MOST principle of stability modulated turbulent exchange.

$$LE = \rho L_{s/f} k^2 z_u z_q \left( \frac{\Delta \bar{u} \Delta \bar{q}}{z^2} \right) (\emptyset_m \emptyset_h)^{-1} ,$$
 (6)

$$H = \rho C_P k^2 z_u z_t \left(\frac{\Delta \bar{u} \Delta \bar{\tau}}{z^2}\right) \left(\emptyset_m \emptyset_q\right)^{-1}, \tag{7}$$

$$\begin{cases} \left(\emptyset_{m}\emptyset_{h,q}\right)^{-1} = (1 - 5Ri_{b})^{2} & (\text{Rib} > 0) \\ \left(\emptyset_{m}\emptyset_{h,q}\right)^{-1} = (1 - 16Ri_{b})^{0.75} (\text{Rib} \le 0), \end{cases}$$
(8)

$$Ri_b = \frac{g_{\overline{\Delta T}}^{\Delta T}}{\bar{\tau}(\frac{\Delta \bar{u}}{2})^2},\tag{9}$$

- where,  $k \sim 0.4$  is the von Kármán constant;  $\bar{u}$ ,  $\bar{q}$ , and  $\bar{T}$  represent the mean wind speed, specific
- humidity, and air temperature, respectively; z is the measurement height, where  $z_{u,t,q}$  are the log mean
- heights defined as:

$$z_{u,t,q} = \frac{z - z_{0u,t,q}}{\ln(\frac{z}{z_{0u,t,q}})},$$
 (10)

- and  $z_{0u,q,t}$  denotes the surface roughness lengths for momentum, humidity, and temperature, which are
- typically assumed equal.

### 183 **3.4.** C<sub>Rib2</sub> method

- Similar to C<sub>Rib1</sub>, the C<sub>Rib2</sub> method is a non-iterative method for calculating turbulent fluxes (Essery
- and Etchevers, 2004). It replaces L and  $\psi$  with  $Ri_b$  and  $f_h(Ri_b)$ , respectively, thereby reducing
- computational complexity while preserving the physical consistency of MOST. Turbulent fluxes in the
- C<sub>Rib2</sub> method are calculated as:

$$LE = \rho L_{s/f} C_H u[q_{sat}(T_s, P_s) - q],$$
 (11)

$$H = \rho C_P C_H u [T_S - T_a],$$
 (12)

- where,  $q_{sat}(T_s, P_s)$  denotes the saturation specific humidity at surface temperature  $T_s$  and pressure  $P_s$ ,
- and  $C_H$  is a surface exchange coefficient. Following Louis (1979), the exchange coefficient for surface

sensible and latent heat flux is calculated as  $C_H = C_{Hn} f_h$ , where

$$C_{Hn} = 0.16 \left[ \ln \left( \frac{z_1}{z_0} \right) \right]^{-2}$$
, (13)

is the neutral exchange coefficient for roughness length  $z_0$  and

$$f_h = \begin{cases} (1 + 10Ri_b)^{-1} & Ri_b \ge 0\\ 1 - 10Ri_b \left(1 + 10Ri_b C_{Hn} \sqrt{-Ri_b} / f_z\right)^{-1} & Ri_b < 0 \end{cases},$$
(14)

with

$$f_z = \frac{1}{4} \left(\frac{z_0}{z_1}\right)^{1/2}$$
, (15)

- Although atmospheric stability is also characterized by  $Ri_b$ , the calculation approach in  $C_{Rib2}$  method
- differs from that in C<sub>Rib1</sub> method as follows:

$$200 Ri_b = \frac{gz_1}{u^2} \left\{ \frac{T_a - T_s}{T_a} + \frac{q - q_{sat}(T_s, P_s)}{q + \epsilon/(1 - \epsilon)} \right\}, (16)$$

where,  $z_1$  is the observation height and  $\epsilon$  is the ratio of the molecular weights of water and dry air.

# 202 **3.5.** C<sub>M-O</sub> method

- This method, which adds stability corrections using universal stability functions, employs a
- complete MOST framework that explicitly calculates the friction velocity ( $u^*$ ),  $\psi$ , and L (Hock and
- Holmgren, 2005). It separately applies roughness lengths for temperature  $(z_{0t})$ , humidity  $(z_{0e})$ , and
- momentum  $(z_{0w})$ . Iterative calculations ensure closure of the turbulent fluxes, utilizing nonlinear stability
- functions from Beljaars and Holtslag (1991) under stable conditions, and Businger-Dyer expressions for
- unstable conditions, iteratively solving for u\* and L (Beljaars and Holtslag, 1991; Paulson, 1970).

$$209 LE = L_{s/f} \frac{0.622\rho}{P_0} \frac{k^2}{\left[\ln\left(\frac{z}{z_{0w}}\right) - \psi_M\left(\frac{z}{L}\right)\right] \left[\ln\left(\frac{z}{z_{0e}}\right) - \psi_E\left(\frac{z}{L}\right)\right]} u(e_a - e_s), (17)$$

$$H = \rho C_P \frac{k^2}{\left[\ln\left(\frac{z}{z_{0W}}\right) - \psi_M\left(\frac{z}{L}\right)\right] \left[\ln\left(\frac{z}{z_{0t}}\right) - \psi_H\left(\frac{z}{L}\right)\right]} u(T_a - T_s), \tag{18}$$

#### 211 3.6. Evaluation of model performance

- We used three statistical measures to quantify model accuracy: root mean square error (RMSE),
- MAD, and mean bias error (MBE). The RMSE quantifies the differences between observed ( $Q_o$ ) and
- modeled  $(Q_m)$  fluxes:

$$RMSE = \sqrt{\frac{1}{n}\sum_{i=1}^{n}(Q_{mi} - Q_{oi})^2},$$
 (19)

The MAD reflects the average magnitude of deviations:

$$MAD = \frac{1}{n} \sum_{i=1}^{n} |Q_{oi} - Q_{mi}|, \qquad (20)$$

The MBE represents systematic biases in model predictions:

MBE =
$$\frac{1}{n} \sum_{i=1}^{n} (Q_{oi} - Q_{mi}),$$
 (21)

#### 220 4 Results

#### 4.1 Meteorological conditions on the Dunde Glacier

Meteorological data observed at 5317 m a.s.l., including air temperature (Ta), glacier surface temperature (Ts), relative humidity (RH), wind speed (u) and direction (WD), and atmospheric pressure (Pres), were collected from 14 May to 12 October, 2023 (Fig. 2). During the study period, daily mean Ta at the Dunde Glacier ranged from -14.1 to 3.3 °C, averaging -4.0 °C (Fig. 2a). Instances of a daily mean  $T_a$  above 0 °C occurred primarily in July and August, which are the warmest months, with a mean value of -1 °C. October was the coldest month, with an average daily mean T<sub>a</sub> of -9.5 °C. Air temperature exhibited pronounced seasonal variability, with higher summer averages (-2.1 °C) compared with spring  $(-7.8 \, ^{\circ}\text{C})$  and autumn  $(-6.5 \, ^{\circ}\text{C})$ . The mean daily T<sub>s</sub> was  $-5.4 \, ^{\circ}\text{C}$ , ranging from  $-17.2 \, ^{\circ}\text{C}$  (7 October) to 0 °C (10 and 11 July, and 14 and 16 August; Fig. 2c). Seasonal variations in T<sub>s</sub> closely mirrored those of T<sub>a</sub>, peaking in summer (average of -3.4 °C) and dropping substantially in spring (average of -9.0 °C) and autumn (average of -8.3 °C). The difference between Ta and Ts also exhibited seasonal variability, reaching a maximum in autumn (1.8 °C), with lower values in spring (1.2 °C) and summer (1.3 °C). In addition, both Ta and Ts displayed clear diurnal cycles, characterized by single-trough and single-peak patterns; T<sub>s</sub> peaked at -1.0 °C around 14:00, while T<sub>a</sub> reached a maximum of -2.0 °C at 16:00, about two hours later. In contrast, both T<sub>s</sub> and T<sub>a</sub> exhibited minimum values at 06:00, with T<sub>s</sub> dropping to -9.4 °C and T<sub>a</sub> to -5.6 °C. The earlier T<sub>s</sub> peak is interpreted to reflect the rapid radiative response of the glacier surface to incoming solar radiation; Ta responds more slowly owing to the thermal inertia of air and gradual development of convective mixing. At night, however, both the glacier surface and air cool synchronously under stable stratification and weak turbulence, resulting in the coincident early morning T<sub>s</sub> and T<sub>a</sub> minima. Generally, T<sub>a</sub> was higher than T<sub>s</sub> for most of a given day, indicative of a persistent

near-surface temperature inversion layer over the glacier surface (Fig. 2b and d).

Mean daily relative humidity at the Dunde Glacier was 56.3%, with notable seasonal variability (Fig. 2e). The highest seasonal average occurred in spring (62.6%), whereas autumn was the driest season (50.7%), particular September, which recorded the lowest monthly average (47.5%). On a diurnal scale, relative humidity exhibited an inverse pattern compared with that of T<sub>a</sub>, characterized by a distinct midday trough and nighttime peak (Fig. 2f). Daily minimum humidity typically occurred around noon, coinciding with peak solar radiation and higher T<sub>a</sub>. Thereafter, relative humidity gradually increased, reaching a maximum of 58.8% at approximately 21:30; this likely resulted from the combined effects of rapid radiative cooling of the glacier surface, low T<sub>a</sub>, and weak wind conditions.

Daily mean wind speeds ranged from 1.7 to 10.5 m s<sup>-1</sup>, averaging 4.1 m s<sup>-1</sup>, with marked seasonal variability (Fig. 2g). The strongest winds occurred during spring (average of 5.51 m s<sup>-1</sup>), while summer recorded the weakest winds (average of 3.6 m s<sup>-1</sup>), particularly August, which recorded the lowest monthly mean (3.1 m s<sup>-1</sup>). Autumn wind speeds were intermediate, averaging 4.6 m s<sup>-1</sup>. On a diurnal scale, wind speed exhibited substantial variation, with an average diurnal range of 1.5 m s<sup>-1</sup>. Morning winds were generally weak, increasing notably after 10:30, peaking around 15:30, and subsequently decreasing to a daily minimum around 23:30 (Fig. 2h). Nighttime wind speeds remained relatively stable, associated with stable atmospheric stratification. Prevailing wind directions were westerly and northwesterly (Fig. 2i and j), highlighting the dominant influence of the mid-latitude westerlies on regional meteorological conditions.

Daily mean atmospheric pressure at the Dunde Glacier during our study period ranged from 527.9 to 538.4 hPa, with an average of 534.4 hPa (Fig. 2k). The highest seasonal mean atmospheric pressure occurred in summer (535.2 hPa) and the lowest in spring (530.6 hPa). On a diurnal scale, atmospheric pressure exhibited a distinct double-peak, double-trough daily pattern, with minimum pressure occurring at 06:00, maximum at 23:30, a secondary minimum between 17:30 and 18:00, and a secondary maximum at 12:00 (Fig. 2l). Variations in atmospheric pressure markedly influence glacier surface energy exchange by altering specific humidity gradients and air density, thus indirectly controlling turbulent fluxes on the glacier surface.

Figure 2. Half-hourly and daily average values (left panels) together with mean diurnal cycles (right panels) of six meteorological variables at the Dunde Glacier for the period May–October, 2023: (a, b) air temperature  $(T_a)$ ; (c, d) surface temperature  $(T_s)$ ; (e, f) relative humidity (RH); (g, h) wind speed (u); (i, j) wind direction (WD); and (k, l) atmospheric pressure (Pres).

## 4.2 Observed turbulent fluxes on the Dunde Glacier

Herein, turbulent fluxes are defined as positive when directed from the atmosphere to the glacier, and negative in the opposite direction. Figure 3 shows that the observed daily mean LE ranged from -35.9 W m<sup>-2</sup> (21 August, 2023) to 10.4 W m<sup>-2</sup> (11 July, 2023), with an average of -9.6 W m<sup>-2</sup> over the study period, indicating persistent sublimation/evaporation from the glacier surface. There was notable seasonal variability, with the absolute mean seasonal LE being highest in spring (-15.1 W m<sup>-2</sup>), lowest in autumn (-8.1 W m<sup>-2</sup>), and moderate in summer (-10.4 W m<sup>-2</sup>), reflecting varying intensities of moisture exchange between the glacier and the atmosphere (Fugger et al., 2022).

In contrast, observed H was generally positive, averaging 6.9 W m $^{-2}$ , indicating continuous heat transfer from the atmosphere to the glacier. There were pronounced seasonal variations, with high mean H values in spring (9.3 W m $^{-2}$ ) and autumn (9.2 W m $^{-2}$ ), and a monthly peak in September (10.1 W m $^{-2}$ ).

The lowest values were recorded in summer, which averaged  $5.4\,\mathrm{W\,m^{-2}}$ , with June having the lowest monthly average (3.1 W m<sup>-2</sup>).

Overall, the absolute magnitude of LE generally exceeded that of H. This resulted in a predominantly negative total turbulent flux (LE + H), averaging -2.9 W m<sup>-2</sup>, indicating net heat loss from the glacier through turbulent exchange. From May to mid-September, the total turbulent flux was negative (glacier losing energy), while from mid-September to mid-October, it became positive (glacier receiving energy). June exhibited the strongest turbulent flux (-8.0 W m<sup>-2</sup>), whereas September recorded the weakest (1.5 W m<sup>-2</sup>). These findings underscore the overall state of heat loss from the glacier through turbulent fluxes during the study period.

Figure 3: Half-hourly (thin lines) and daily average (thick lines) variations in (a) latent heat flux, (b) sensible heat flux, and (c) total turbulent flux over the Dunde Glacier during the study period. The green line in (c) represents the daily average of total turbulent flux.

# 4.3 Evaluation of different turbulent flux parameterizations

In this section, we present the results of our evaluation of the simulation ability of five turbulent flux methods ( $C_{kat}$ ,  $C_{log}$ ,  $C_{Rib1}$ ,  $C_{Rib2}$ , and  $C_{M-O}$ ) across different timescales.

## 4.3.1 Overall model performance during the entire study period

We compared the temporal evolution of modeled LE and H at the Dunde Glacier against in-situ observations (Fig. 4). Over the observation period, all models captured the general trend of turbulent fluxes reasonably well. However, during May, when turbulent fluxes fluctuated to the greatest degree, all model simulations substantially underestimated the observed variability. Model-simulated H also deviated considerably from observations in early to mid-September, characterized by systematic underestimation. Additionally, during abrupt changes in observed LE in late June and mid-August, the C<sub>kat</sub> and C<sub>log</sub> methods failed to reproduce the observed flux transitions, markedly underestimating LE during these periods.

Figure 4: Time series of daily mean latent heat (LE) and sensible heat (H) fluxes modeled by the five different schemes ( $C_{kat}$ ,  $C_{log}$ ,  $C_{Rib1}$ ,  $C_{Rib2}$ , and  $C_{M-O}$ ) compared with eddy covariance-derived values (measured).

Statistical comparisons between modeled and observed fluxes were conducted for the entire study period to quantitatively assess method performance (Table 2). For LE,  $C_{kat}$  and  $C_{M-O}$  method slightly underestimated the flux, while the other three methods ( $C_{log}$ ,  $C_{Rib1}$  and  $C_{Rib2}$ ) tended to give overestimations. The  $C_{kat}$  method exhibited the highest RMSE and MAD values, indicating poorer LE simulation accuracy compared with the other methods. Meanwhile,  $C_{log}$  presented the largest systematic bias (MBE = 1.96). Hence, the  $C_{Rib1}$ ,  $C_{Rib2}$ , and  $C_{M-O}$  methods performed best in terms of LE, with the  $C_{Rib1}$  and  $C_{Rib2}$  methods slightly underestimating the flux, and the  $C_{M-O}$  method slightly overestimating it. For H, all methods showed some underestimation, with the  $C_{kat}$  and  $C_{M-O}$  methods exhibiting lower RMSE values (8.60 and 8.63 W m<sup>-2</sup>, respectively), indicating the best accuracy among the tested methods.

Table 2: Comparative statistics of modeled latent heat (LE) and sensible heat (H) fluxes over the entire study period (based on adjusted parameterizations) against eddy covariance observations. RMSE = root mean square error; MAD = mean absolute deviation; MBE = mean bias error.

|                  |      | LE (W m <sup>-2</sup> ) |       |      | H (W m <sup>-2</sup> ) |       |
|------------------|------|-------------------------|-------|------|------------------------|-------|
| Method           | RMSE | MAD                     | MBE   | RMSE | MAD                    | MBE   |
| C <sub>kat</sub> | 7.96 | 6.12                    | -0.31 | 8.60 | 6.02                   | -3.83 |
| $C_{log}$        | 6.82 | 4.82                    | 1.96  | 8.82 | 6.08                   | -5.17 |
| $C_{Rib1}$       | 6.62 | 4.78                    | 0.03  | 8.80 | 6.43                   | -5.16 |
| $C_{Rib2}$       | 6.35 | 4.48                    | 0.30  | 8.85 | 6.22                   | -5.58 |
| $C_{M\!-\!O}$    | 6.36 | 4.64                    | -0.66 | 8.63 | 6.18                   | -5.46 |

4.3.2 Seasonal variability in method performance

To further explore method behavior under different atmospheric conditions, we conducted seasonal evaluations of the five turbulent flux methods, providing critical insights into their applicability across varying meteorological regimes (Table 3). All five methods exhibited similar patterns of seasonal error. For LE, the largest simulation errors occurred in spring, typically manifested as underestimations, while the smallest errors were found in autumn, with a tendency for slight overestimations. For H, the largest errors occurred in spring and the smallest in summer, with underestimations in both cases. These findings suggest that the simulation accuracy of all methods decreases under conditions of intense turbulent exchange, such as high temperature, humidity, and wind speed, whereas performance is more stable

during other seasons.

In a seasonal context, for LE, all parameterizations consistently exhibited the lowest simulation accuracy in spring (Table 3). The C<sub>M-O</sub> scheme achieved the best spring performance, with an RMSE of 8.48 W m<sup>-2</sup> and MAD of 6.59 W m<sup>-2</sup>, these values being markedly lower than those of the other schemes. In summer, the C<sub>Rib2</sub> and C<sub>M-O</sub> methods exhibited the lowest RMSE values, outperforming the other schemes. In autumn, the C<sub>log</sub> method yielded the best LE simulation performance (RMSE: 3.86 W m<sup>-2</sup>, MAD: 2.68 W m<sup>-2</sup>), while the C<sub>kat</sub> method exhibited the lowest accuracy. The remaining schemes displayed intermediate and comparable performances. In addition, the C<sub>kat</sub> and C<sub>log</sub> methods exhibited the greatest seasonal variability in accuracy, with RMSE differences between spring and autumn reaching 6.84 and 6.70 W m<sup>-2</sup>, respectively, indicating poor seasonal consistency and limited robustness under non-extreme conditions. In contrast, the C<sub>M-O</sub> method demonstrated remarkable seasonal stability, with only a 3.26 W m<sup>-2</sup> difference in RMSE between spring and autumn, highlighting its adaptability and robustness across seasons.

For H, all methods exhibited reduced accuracy in spring, although the C<sub>kat</sub> method performed relatively well with a RMSE of 13.78 W m<sup>-2</sup>, MAD of 8.65 W m<sup>-2</sup>, and MBE of -6.3 W m<sup>-2</sup>. Other methods showed similar spring statistics. Summer was the best-performing season for H across all parameterizations, with the C<sub>M-O</sub> method achieving the lowest summer errors. In autumn, both C<sub>kat</sub> and C<sub>M-O</sub> yielded relatively low RMSE values (8.71 and 8.73 W m<sup>-2</sup>, respectively), while the C<sub>Rib2</sub> method exhibited the highest RMSE at 9.19 W m<sup>-2</sup>; the remaining schemes produced intermediate and comparable results. Overall, the C<sub>M-O</sub> method achieved the best performance for H during both summer and autumn.

Table 3: Seasonal performance of latent heat (LE) and sensible heat (H) flux methods, evaluated against eddy covariance observations. RMSE = root mean square error; MAD = mean absolute deviation; MBE = mean bias error.

|                  |        | :     | LE (W m <sup>-2</sup> ) |       |       | H (W m <sup>-2</sup> ) |       |
|------------------|--------|-------|-------------------------|-------|-------|------------------------|-------|
| Method           | Season | RMSE  | MAD                     | MBE   | RMSE  | MAD                    | MBE   |
|                  | Spring | 12.78 | 9.66                    | 7.02  | 13.78 | 8.65                   | -6.30 |
| $C_{\text{kat}}$ | Summer | 7.47  | 5.99                    | -1.13 | 7.07  | 5.17                   | -2.61 |
|                  | Autumn | 5.94  | 4.81                    | -1.76 | 8.71  | 6.73                   | -5.41 |

|                   | Spring | 10.56 | 8.10 | 6.39  | 14.19 | 9.18  | -7.77 |
|-------------------|--------|-------|------|-------|-------|-------|-------|
| $C_{\text{log}}$  | Summer | 6.86  | 5.07 | 2.21  | 7.13  | 5.01  | -3.98 |
|                   | Autumn | 3.86  | 2.68 | -0.65 | 9.12  | 7.08  | -6.64 |
|                   | Spring | 9.41  | 7.06 | 4.20  | 14.79 | 10.71 | -9.67 |
| $C_{Rib1}$        | Summer | 6.49  | 4.71 | 0.41  | 7.01  | 5.42  | -3.79 |
|                   | Autumn | 5.19  | 3.87 | -2.79 | 8.83  | 6.79  | -6.19 |
| C <sub>Rib2</sub> | Spring | 9.37  | 7.05 | 4.61  | 14.48 | 9.85  | -8.88 |
|                   | Summer | 6.22  | 4.49 | 0.66  | 7.04  | 5.06  | -4.32 |
|                   | Autumn | 4.70  | 3.29 | -2.55 | 9.19  | 7.16  | -6.90 |
| См-о              | Spring | 8.48  | 6.59 | 2.92  | 14.81 | 10.71 | -9.88 |
|                   | Summer | 6.32  | 4.55 | -0.14 | 6.70  | 5.01  | -4.26 |
|                   | Autumn | 5.22  | 3.94 | -3.51 | 8.73  | 6.78  | -6.16 |

4.3.3 Evaluation of diurnal variations

An evaluation of diurnal variations in modeled turbulent fluxes offers important diagnostic information on the simulation ability of a given parameterization. At the diurnal scale, the simulated turbulent fluxes aligned well with in-situ observations, with both LE and H exhibiting a distinct single-trough daily pattern (Fig. 5). For LE, the simulated values obtained from all methods exhibited higher accuracy during nighttime and early morning hours, with smaller deviations from observed values (Fig. 5a). In contrast, during periods of intense flux variability—typically around noon—method performance declined, and simulation errors became larger. For H, the methods generally performed better than for LE during noon, but all exhibited limited capability in capturing the early-morning fluctuations (Fig. 5b). Among the five schemes, the CM-O method exhibited the best overall performance (Fig. 5); it accurately captured turbulent fluxes during the relatively stable 18:00–08:00 period, and despite some degree of error, it most effectively reproduced the pronounced fluctuations in LE and H during the period of intensified flux activity (13:00–15:00).

Figure 5: Mean diurnal cycles of modeled latent heat and sensible heat fluxes obtained from the five different schemes ( $C_{kat}$ ,  $C_{log}$ ,  $C_{Rib1}$ ,  $C_{Rib2}$ , and  $C_{M-O}$ ) compared with eddy covariance-derived values ( $LE_{EC}$  and  $H_{EC}$ ).

In summary, among the five evaluated methods, the  $C_{M-O}$  method delivered the most accurate and stable performance for LE throughout the study period, and it also ranked among the top performers for H. Both the  $C_{kat}$  and  $C_{M-O}$  methods performed well for H, without any substantial seasonal deficiencies. Moreover, the  $C_{M-O}$  method also demonstrated the best agreement with observations in reproducing the diurnal variations in LE and H across different seasons. Therefore, the  $C_{M-O}$  method emerges as a particularly suitable option for simulating turbulent fluxes on the Dunde Glacier, offering both high accuracy and temporal robustness on daily and seasonal scales.

### 5 Discussion

#### 5.1 Differences in performance among turbulent flux methods

Variations in the performance of different turbulent flux parameterizations primarily arise from their respective approaches to computing turbulent exchange coefficients and applying atmospheric stability corrections. These methodological differences directly affect method accuracy under varying temperature gradients, humidity gradients, and wind speed conditions, leading to discrepancies in the simulated LE and H.

The  $C_{kat}$  method explicitly considers katabatic flows, introducing the katabatic bulk exchange coefficient ( $C_{kat}$ ) to partially overcome the limitations of MOST during strong katabatic wind conditions (Oerlemans and Grisogono, 2002). This approach emphasizes intensified katabatic turbulence during nocturnal cooling or cold-air intrusions, enhancing turbulent exchange (Horst and Doran, 1988). However, the  $C_{kat}$  method lacks explicit stability corrections and dynamic roughness length adjustment,

application is more suited to small and medium-sized glaciers rather than large-scale domains such as 407 the Greenland Ice Sheet. 408 The  $C_{log}$  method employs a constant exchange coefficient ( $C_H$ ) driven by near-surface wind speed 409 and the difference in air temperature (or humidity) and surface temperature (or humidity), providing a 410 simplified, computationally efficient structure suitable for large-scale climate models (Oerlemans, 2000). 411 Being a wind-speed-driven scheme, it performs well under less stable stratification conditions. However, 412 the method also exhibits certain limitations. Owing to the absence of atmospheric stability parameters in 413 its structure, it cannot identify or respond to surface inversions forming at night or in the early morning. 414 Herein, the LE RMSE in spring reached 10.56 W m<sup>-2</sup>, ranking among the largest errors of all evaluated 415 methods (Table 3); this method also exhibited substantial diurnal simulation errors. 416 The CRib1 and CRib2 methods are based on a single-step turbulence parameterization method centered 417 on the Richardson number for stability correction. Their structures are concise, do not rely on multi-level 418 observational data, and avoid complex iterative procedures, making them among the most robust 419 parameterization schemes for simulating turbulent fluxes. However, these approaches also have 420 limitations. They do not dynamically iterate L, which limits their responsiveness to complex and rapidly 421 changing atmospheric conditions when compared with fully dynamic MOST-based methods (Högström, 422 1988). In addition, the adjustment of  $z_0$  not only affects the simulation of turbulent fluxes but also 423 influences surface albedo, complicating independent parameter calibration and hindering reliable flux 424 validation. 425 The C<sub>M-O</sub> method, which exhibited the best performance among all evaluated methods in this study, 426 adopts a complete implementation of the MOST framework, explicitly calculating u\*, stability correction 427 functions, and L. The scheme separately applies distinct roughness lengths for momentum  $(z_{0w})$ , 428 temperature  $(z_{0t})$ , and humidity  $(z_{0e})$ , enhancing physical consistency in flux estimation. The utilization 429 of a robust iterative algorithm ensures energy balance closure. From our overall statistical evaluation, the 430 C<sub>M-O</sub> method achieved one of the best performances among the five evaluated methods, with an RMSE 431 of only 6.36 W m<sup>-2</sup> for LE and 8.63 W m<sup>-2</sup> for H (Table 2). More importantly, the C<sub>M-O</sub> method exhibited 432 minimal seasonal variability in flux estimation accuracy. The RMSE values for LE in spring, summer, 433 and autumn were 8.48, 6.32, and 5.22 W m<sup>-2</sup>, respectively, giving an inter-seasonal RMSE difference of 434 approximately 3.3 W m<sup>-2</sup>, much smaller than the differences observed in other methods (e.g., a difference

leading to a slow response under rapidly changing surface conditions (Denby, 2000). Hence, its

https://doi.org/10.5194/egusphere-2025-4227 Preprint. Discussion started: 15 October 2025 © Author(s) 2025. CC BY 4.0 License.

exceeding 5 W m<sup>-2</sup> for the  $C_{Rib1}$  method). Additionally, the  $C_{M-O}$  method demonstrated higher sensitivity and precision in simulating diurnal variations in turbulent fluxes, showing the closest alignment with observed diurnal cycles of LE and H, particularly during periods of intense midday convection. Nonetheless, the  $C_{M-O}$  method also possesses structural limitations. Specifically, the roughness lengths  $(z_{0t}, z_{0e}, \text{ and } z_{0w})$  lack direct observational support, necessitating iterative parameter calibration, thereby reducing method interpretability and generalizability.

# 5.2 Performance improvement of the SEB model by optimizing turbulent flux methods

To assess the importance of turbulent flux methods in simulating glacier mass balance, we included three representative turbulent flux methods (the C<sub>log</sub>, C<sub>Rib2</sub>, and C<sub>M-O</sub> methods) in a coupled energy balance–snow and firn model (EBFM; Van Pelt et al., 2012, 2019) applied at the point scale. The turbulent fluxes, T<sub>s</sub>, ice surface height, and cumulative mass balance (CMB) were simulated using both original and recalibrated parameters, and compared against observational data (Fig. 6). Quantitative evaluation indices, including the correlation coefficient (R), RMSE, and MAD and estimates of CMB, were used to demonstrate improvements resulting from model optimization (Table 4).

Figure 6: Time series of turbulent fluxes, cumulative mass balance (CMB), surface temperature, and surface height over the Dunde Glacier during the study period, based on original methods (before), recalibrated methods (after), and observations (obs).

Table 4: Comparative statistics between observations and surface energy balance model simulations of surface temperature (T<sub>s</sub>), surface height, and cumulative mass balance at the Dunde Glacier, using both original (before) and recalibrated (after) parameters.

|                   | Before                  |                                     |              |                         | After                                 |              |  |  |
|-------------------|-------------------------|-------------------------------------|--------------|-------------------------|---------------------------------------|--------------|--|--|
| Method            | $T_s$                   | Surface height                      | Mass balance | Ts                      | Surface height                        | Mass balance |  |  |
| Clog              | R = 0.84<br>RMSE = 3.26 | R = 0.93 $RMSE = 0.16$ $MAD = 0.14$ | -238.41      | R = 0.85<br>RMSE = 2.45 | R = 0.93<br>RMSE = 0.15<br>MAD = 0.14 | -262.14      |  |  |
| C <sub>Rib2</sub> | R = 0.78<br>RMSE = 8.79 | R = 0.91 $RMSE = 0.18$ $MAD = 0.16$ | -225.77      | R = 0.85<br>RMSE = 2.53 | R = 0.93<br>RMSE = 0.15<br>MAD = 0.13 | -252.87      |  |  |
| См-о              | R = 0.86<br>RMSE = 3.16 | R = 0.94 $RMSE = 0.16$ $MAD = 0.14$ | -242.41      | R = 0.86<br>RMSE = 2.53 | R = 0.94 $RMSE = 0.14$ $MAD = 0.13$   | -256.80      |  |  |

https://doi.org/10.5194/egusphere-2025-4227 Preprint. Discussion started: 15 October 2025 © Author(s) 2025. CC BY 4.0 License.

We first ran the SEB models under uncalibrated conditions with three turbulent flux methods (the Clog, CRib2, and CM-O methods). These gave RMSE values for Ts of 3.26, 8.79, and 2.53 °C, respectively. The R values between modeled and measured Ts exceeded 0.78 for all three schemes. The RMSE values for surface height change ranged from 0.16 to 0.18 m, while the CMB estimates were -238.41, -225.77, and -242.41 mm, respectively, all deviating markedly from the observed value of -304.19 mm. Following parameter calibration, all three schemes exhibited marked improvements in model performance. The CRib2 method showed the greatest improvement, giving a CMB value of -252.87 mm, equating to an error reduction of 34.55% relative to the uncalibrated result. In parallel, the RMSE values for Ts and surface height decreased by 71.2% and 16.6%, respectively. The CM-O method-based model yielded the highest overall accuracy after optimization, with R values of 0.86 for Ts and 0.94 for surface height. Although a residual bias remained in the CMB simulation, it was substantially mitigated compared with the original configuration. Overall, these findings highlight that turbulent flux parameter optimization is an effective approach to improve glacier SEB model accuracy.

# 5.3 Evaluation of turbulent flux methods under extreme weather conditions

Our analysis above demonstrates that the optimization of turbulent flux parameterizations markedly enhances the accuracy of glacier surface energy and mass balance simulations under the mean climate state (e.g., multi-year averages of Ta and precipitation). However, the ability of SEB models to simulate turbulent fluxes under extreme weather and climate events remains uncertain and warrants further investigation. Between 8 and 11 July, 2023, the Dunde Glacier experienced a typical humid heatwave (Fig. 7). During this event, the daily average air temperature reached 2.4 °C, this being substantially higher than the July average outside of the heatwave event (-1.5 °C). Nighttime air temperatures from 9 to 11 July consistently exceeded 0 °C, exhibiting reduced diurnal fluctuations, and then sharply dropped below freezing on 12 July immediately after the heatwave event. Relative humidity during the humid heatwave reached an average of 94.3%, this being markedly higher than during the July non-heatwave period (89.1%). The concurrent anomalies in air temperature and humidity during this humid heatwave would inevitably lead to anomalous turbulent fluxes.

Figure 7: Half-hourly values of (a) air temperature  $(T_a)$ , (b) relative humidity (RH), (c) latent heat flux (LE), and (d) sensible heat flux (H) over the Dunde Glacier during July, 2023. The yellow-shaded area highlights the period of the humid heatwave event (8–11 July, 2023).

To clarify turbulent flux behaviors under such anomalous climatic conditions, we compared turbulent fluxes during the humid heatwave (8–11 July, 2023) with those during the non-heatwave period (the remainder of July, 2023) based on observational data. The H markedly increased during the humid heatwave, reaching a daily average of 18.3 W m<sup>-2</sup>, this being nearly three times higher than the non-heatwave average (6.13 W m<sup>-2</sup>). The peak H of 128.2 W m<sup>-2</sup> occurred at 15:00 on 11 July, 2023. Moreover, the diurnal variation in H during the heatwave event ranged from 0.85 to 61.2 W m<sup>-2</sup>, substantially exceeding the range observed during the non-heatwave period (–5.0 to 16.3 W m<sup>-2</sup>). This extraordinary increase in H during the humid heatwave indicates that substantial heat was transferred from the atmosphere to the glacier during this event. Concurrently, LE exhibited a remarkable reversal during the humid heatwave. During the non-heatwave period, LE remained negative, with an average value of –12.5 W m<sup>-2</sup>. Such negative LE values are a common characteristic of continental glaciers on the TP. However, during the humid heatwave, LE values became positive, averaging 0.11 W m<sup>-2</sup>. This shift was further

509510

reflected in diurnal LE variations, with a range of -19.2 to 24.3 W m<sup>-2</sup> during the humid heatwave, which was markedly different from the range of -36.5 to 1.2 W m<sup>-2</sup> observed during the rest of July. This unusual reversal implies pronounced atmosphere-to-surface moisture condensation, resulting in increased heat input onto the glacier surface (Zhang et al., 2017). Such positive LE values are commonly observed over maritime glaciers (Yang et al., 2011). The glacier surface transitioned into a state of net turbulent energy gain, with the mean total turbulent flux (LE + H) reaching  $18.53~\mathrm{W}~m^{-2}$  during the humid heatwave. In contrast, the non-heatwave period in July was characterized by net turbulent energy loss, averaging -6.25 W m<sup>-2</sup>. This contrast underscores a distinct shift from turbulent heat loss to heat gain under humid heatwave conditions. Concurrently, surface albedo decreased and melt energy increased, triggering intense ablation. Thus, although short-lived, such anomalous turbulent flux variations during the humid heatwave markedly altered the glacier's energy-balance processes by modifying albedo and melt energy, with potential implications for longer-term melt dynamics (Zhu et al., 2024b). We further evaluated the ability of the five methods to simulate turbulent fluxes under extreme weather and climate events. Among the five turbulent flux methods tested, all captured the pronounced increase in H during the humid heatwave (Fig. 4), with the C<sub>M-O</sub> method showing the best performance. Although the C<sub>M-O</sub> method calculated mean (10.2 W m<sup>-2</sup>) was lower than the observed value (18.3 W m<sup>-2</sup>), it outperformed the other schemes. The C<sub>log</sub> method exhibited the poorest performance, simulating a mean H of only 5.3 W m<sup>-2</sup>. Meanwhile, the mean H values during the humid heatwave simulated by the Ckat, CRib1, and CRib2 methods were 5.7, 8.9, and 5.5 W m<sup>-2</sup>, respectively. In contrast, all methods failed to reproduce the observed reversal of LE during the humid heatwave (Fig. 4). The mean LE values simulated by the  $C_{kat}$ ,  $C_{log}$ ,  $C_{Rib1}$ ,  $C_{Rib2}$ , and  $C_{M-0}$  methods were -5.8, -6.2, -6.7, -6.9, and -7.7 W m<sup>-2</sup>, respectively, all substantially deviating from the observed value of 0.11 W m<sup>-2</sup>. The reasons behind this discrepancy warrant further investigation. Overall, the methods, particularly the C<sub>M-O</sub> method, demonstrated a strong response to the anomalous increase in H during the humid heatwave. However, none of the methods captured the observed reversal of LE. All calculated values were markedly lower than the measured values, suggesting that current turbulent flux parameterizations may underestimate the magnitudes of H and LE under extreme meteorological conditions, and hence underestimate their contributions to glacier melting.

Under ongoing global warming, the climate system is recording notable warming and moistening

trends on the TP (Wang et al., 2010). This means that the TP will likely experience an increased frequency and intensity of humid heatwave events (Zhang et al., 2025). Such conditions will cause continental glaciers, such as the Dunde Glacier, to increasingly exhibit turbulent flux characteristics that resemble those of maritime glaciers, including reversed LE directions and markedly increased H. Thus, continued warming may drive future glacier-type transitions.

## 6. Conclusions

In this study, we conducted a comprehensive analysis of turbulent fluxes on the continental Dunde Glacier (Qilian Mountains, TP), using continuous meteorological observations and EC flux measurements collected from May to October, 2023. Throughout the observation period, the Dunde Glacier exhibited energy loss through LE (mean =  $-10.34 \text{ W m}^{-2}$ ) while simultaneously gaining energy via H from the atmosphere (mean =  $6.93 \text{ W m}^{-2}$ ). Clear seasonal variations in turbulent fluxes were observed: the absolute magnitude of LE was largest in spring (mean  $-15.1 \text{ W m}^{-2}$ ) and smallest in autumn (mean  $-8.1 \text{ W m}^{-2}$ ), whereas H was higher during spring and autumn (means  $9.3 \text{ and } 9.2 \text{ W m}^{-2}$ , respectively) and lower in summer (mean  $5.4 \text{ W m}^{-2}$ ). The magnitude of LE exceeded that of H, resulting in continuous net heat loss from the glacier surface via turbulent exchange.

In addition, we systematically assessed the performance of commonly used turbulent flux methods. Among the five schemes evaluated, the method based on Monin–Obukhov similarity theory with universal stability functions demonstrated the best overall performance in simulating turbulent fluxes, both in terms of simulation accuracy (RMSE of 6.36 W m<sup>-2</sup> for LE and 8.63 W m<sup>-2</sup> for H) and in capturing diurnal variability. Furthermore, after parameter optimization, the SEB model incorporating this method achieved the highest accuracy, with R values of 0.86 and 0.94 for T<sub>s</sub> and ice surface height, respectively. These results suggest that the method based on Monin–Obukhov similarity theory with universal stability functions may be the most suitable for estimating turbulent fluxes over continental glaciers on the TP.

Lastly, we investigated the impacts of extreme weather and climate events (e.g., humid heatwaves) on turbulent flux characteristics. The analyzed humid heatwave event markedly altered the typical turbulent flux patterns, resulting in a dramatic increase in H and reversal of the LE direction; this caused the average turbulent flux to reach 18.53 W m<sup>-2</sup> during the humid heatwave period, temporarily transitioning the glacier surface from a heat-loss to a pronounced heat-gain state. However, all methods

exhibited increased errors under high-temperature, high-humidity, and intensely turbulent conditions (e.g., during spring or heatwave periods), demonstrating substantial limitations in method robustness under extreme or rapidly changing climate conditions. During the analyzed humid heatwave, all methods, and particularly the C<sub>M-O</sub> method, a complete implementation of the Monin-Obukhov similarity theory framework, successfully captured the sharp increase in H, although the simulated values were consistently lower than the observed values. Meanwhile, none of the methods reproduced the reversal of LE direction, indicating that the contribution of turbulent fluxes to glacier melt energy under extreme meteorological conditions is likely underestimated. In summary, our findings emphasize the critical importance of turbulent flux parameterizations for accurate glacier SEB modeling and specifically highlight the superior performance and applicability of the method based on Monin-Obukhov similarity theory with universal stability functions in mid-latitude continental glacier environments. Future research should intensify efforts to elucidate the physical mechanisms behind anomalous turbulent flux behavior during extreme weather and climate events (e.g., humid heatwaves), and thus enhance modeling accuracy under exceptional meteorological conditions. Acknowledgments We gratefully acknowledge the Forestry and Grassland Bureau of Delingha City, Qinghai Province, for their support during our field observations. We also thank the Science Observation Station Management Center of Lanzhou University for providing part of the observation instruments. Code/Data Availability ERA5 is available for download at https://cds.climate.copernicus.eu/cdsa pp#!/search?type=dataset. The EBFM model is available via https://github.com/wardvp/EBFM-gla cier. Data supporting the findings of this study are available from the corresponding author up on reasonable request. Author contribution XYC: data curation, formal analysis, visualization, writing (original draft preparation). ZML: conceptualization, methodology, supervision, writing (review and editing), field observations, funding acquisition. GJ: conceptualization, supervision, writing (review and editing), field observations, project administration. LDJ: data curation, field observations, writing (review and editing). HL: data curation, resources. ZF: data curation. ZHB: field observations, data curation. ZFY: project administration. Competing interests The authors declare that they have no conflict of interest. Financial support This study was jointly supported by the National Key R & D Program of China (Grant No. 2024YFF0808601), the Second Tibetan Plateau Scientific Expedition and Research Program (Grant

No. 2024QZKK0400), the National Natural Science Foundation of China (Grant No. 42471142), and 592 Science and Technology Program of Gansu Province (Grant No. 24YFFF002). 593 References 594 Andreas, E.: A theory for the scalar roughness and the scalar transfer-coefficients over snow and sea ice, 595 Bound.-Lay. Meteorol., 38, 159-184, https://doi.org/10.1007/BF00121562, 1987. 596 Azam, M. F., Wagnon, P., Vincent, C., Ramanathan, A., Favier, V., Mandal, A., and Pottakkal, J. G.: 597 Processes governing the mass balance of Chhota Shigri Glacier (western Himalaya, India) assessed 598 by point-scale surface energy balance measurements, The Cryosphere, 8, 2195-2217, https://doi.org/10.5194/tc-8-2195-2014, 2014. 599 600 Beljaars, A. C. M. and Holtslag, A. A. M.: Flux Parameterization over Land Surfaces for Atmospheric 601 Models, 327-341, https://doi.org/10.1175/1520-Appl. Meteorol., 602 0450(1991)030<0327:FPOLSF>2.0.CO;2, 1991. 603 Box, J. E. and Steffen, K.: Sublimation on the Greenland Ice Sheet from automated weather station observations, J. Geophys. Res.: Atmos., 106, 33965–33981, https://doi.org/10.1029/2001JD900219, 604 605 2001. 606 Brun, F., Berthier, E., Wagnon, P., Kaab, A., and Treichler, D.: A spatially resolved estimate of High 607 Mountain Asia glacier mass balances from 2000 to 2016 (vol 10, pg 668, 2017), Nat. Geosci., 11, 608 543-543, https://doi.org/10.1038/s41561-018-0171-z, 2018. 609 Cullen, N. J., Mölg, T., Kaser, G., Steffen, K., and Hardy, D. R.: Energy-balance model validation on the 610 top of Kilimanjaro, Tanzania, using eddy covariance data, Ann. Glaciol., 46, 227-233, 611 https://doi.org/10.3189/172756407782871224, 2007. Denby, B.: The use of bulk and profile methods for determining surface heat fluxes in the presence of 612 613 glacier winds, J. Glaciol., 46, 445-452, https://doi.org/10.3189/172756500781833124, 2000. 614 Duan, K., Yao, T., Wang, N., and Liu, H.: Numerical simulation of Urumqi Glacier No. 1 in the eastern 615 central Asia from 2005 to 2070, Chin. Sci. Bull., 57, 4505-4509, 616 https://doi.org/10.1007/s11434-012-5469-4, 2012. 617 Dyer, A. J.: A review of flux-profile relationships, Bound.-Lay. Meteorol., 7, 363-372, https://doi.org/10.1007/BF00240838, 1974. 618

109, 2004JD005036, https://doi.org/10.1029/2004JD005036, 2004. 620 621 Fugger, S., Fyffe, C. L., Fatichi, S., Miles, E., McCarthy, M., Shaw, T. E., Ding, B., Yang, W., Wagnon, 622 P., Immerzeel, W., Liu, Q., and Pellicciotti, F.: Understanding monsoon controls on the energy and 623 mass balance of glaciers in the Central and Eastern Himalaya, The Cryosphere, 16, 1631-1652, 624 https://doi.org/10.5194/tc-16-1631-2022, 2022. 625 Gao, J., Yao, T., Masson-Delmotte, V., Steen-Larsen, H. C., and Wang, W.: Collapsing glaciers threaten 626 Asia's water supplies, Nature, 565, 19–21, https://doi.org/10.1038/d41586-018-07838-4, 2019. 627 Guo, W., Liu, S., Xu, L., Wu, L., Shangguan, D., Yao, X., Wei, J., Bao, W., Yu, P., Liu, Q., and Jiang, Z.: 628 The second Chinese glacier inventory: data, methods and results, J. Glaciol., 61, 357-372, 629 https://doi.org/10.3189/2015JoG14J209, 2015. 630 Guo, X., Yang, K., Zhao, L., Yang, W., Li, S., Zhu, M., Yao, T., and Chen, Y.: Critical Evaluation of 631 Scalar Roughness Length Parametrizations Over a Melting Valley Glacier, Bound.-Lay. Meteorol., 139, 307-332, https://doi.org/10.1007/s10546-010-9586-9, 2011. 632 633 Hock, R. and Holmgren, B.: A distributed surface energy-balance model for complex topography and its 634 application Storglaciären, Sweden, J. Glaciol., 51, 25-36, 635 https://doi.org/10.3189/172756505781829566, 2005. 636 Högström, U.: Non-dimensional wind and temperature profiles in the atmospheric surface layer: A re-637 evaluation, Bound.-Lay. Meteorol., 42, 55-78, https://doi.org/10.1007/BF00119875, 1988. 638 Holtslag, A. A. M. and de Bruin, H. A. R.: Applied Modeling of the Nighttime Surface Energy Balance 639 over Land, J. Appl. Meteorol. Climatol., 27, 689-704, <a href="https://doi.org/10.1175/1520-">https://doi.org/10.1175/1520-</a> 0450(1988)027<0689:AMOTNS>2.0.CO;2, 1988. 640 641 Horst, T. and Doran, J.: The Turbulence Structure of Nocturnal Slope Flow, J. Atmos. Sci., 45, 605-616, 642 https://doi.org/10.1175/1520-0469(1988)045<0605:TTSONS>2.0.CO;2, 1988. 643 Hugonnet, R., McNabb, R., Berthier, E., Menounos, B., Nuth, C., Girod, L., Farinotti, D., Huss, M., 644 Dussaillant, I., Brun, F., and Kaab, A.: Accelerated global glacier mass loss in the early twenty-first 645 century, Nature, 592, 726-731, https://doi.org/10.1038/s41586-021-03436-z, 2021. 646 Kuipers Munneke, P., van den Broeke, M. R., Reijmer, C. H., Helsen, M. M., Boot, W., Schneebeli, M., 647 and Steffen, K.: The role of radiation penetration in the energy budget of the snowpack at Summit, Greenland, The Cryosphere, 3, 155–165, https://doi.org/10.5194/tc-3-155-2009, 2009. 648

Essery, R. and Etchevers, P.: Parameter sensitivity in simulations of snowmelt, J. Geophys. Res.: Atmos.,

649 Li, C., Zhou, L., Qin, D., Liu, L., Qin, X., Wang, Z., and Ren, J.: Preliminary study of atmospheric carbon 650 dioxide in a glacial area of the Qilian Mountains, west China, Atmos. Environ., 99, 485-490, 651 https://doi.org/10.1016/j.atmosenv.2014.10.020, 2014. Li, J., Liu, S., Zhang, Y., and Shangguan, D.: Surface energy balance of Keqicar Glacier, Tianshan 652 653 Mountains, China, during ablation period, Sci. Cold Arid Reg., 3, 654 doi:10.3724/SP.J.1226.2011.00197, 2011. 655 Louis, J.-F.: A parametric model of vertical eddy fluxes in the atmosphere, Bound.-Lay. Meteorol., 17, 187-202, https://doi.org/10.1007/BF00117978, 1979. 656 657 Mandal, A., Angchuk, T., Azam, M. F., Ramanathan, A., Wagnon, P., Soheb, M., and Singh, C.: An 11year record of wintertime snow-surface energy balance and sublimation at 4863 m a.s.l. on the 658 659 Chhota Shigri Glacier moraine (western Himalaya, India), The Cryosphere, 16, 3775-3799, https://doi.org/10.5194/tc-16-3775-2022, 2022. 660 661 Mölg, T., Maussion, F., Yang, W., and Scherer, D.: The footprint of Asian monsoon dynamics in the mass 662 and energy balance of a Tibetan glacier, The Cryosphere, 6, 1445-1461, https://doi.org/10.5194/tc-663 6-1445-2012, 2012. 664 Monin, A. S. and Obukhov, A. M.: Basic laws of turbulent mixing in the surface layer of the atmosphere, 665 Tr. Akad. Nauk SSSR Geophiz. Inst., 24(151), 163-187, 1954. 666 Oerlemans, J.: Analysis of a 3 year meteorological record from the ablation zone of Morteratschgletscher, 667 Switzerland: Energy balance, Glaciol., 46, 571-579, and mass 668 https://doi.org/10.3189/172756500781832657, 2000. 669 Oerlemans, J. and Grisogono, B.: Glacier winds and parameterisation of the related surface heat fluxes, 670 Tellus A, 54, 440-452, https://doi.org/10.1034/j.1600-0870.2002.201398.x, 2002. 671 Paulson, C. A.: The Mathematical Representation of Wind Speed and Temperature Profiles in the 672 Unstable Atmospheric Surface Layer, J. Appl. Meteorol. Climatol., 9, 857-861, https://doi.org/10.1175/1520-0450(1970)009<0857:TMROWS>2.0.CO;2, 1970. 673 674 Potocki, M., Mayewski, P. A., Matthews, T., Perry, L. B., Schwikowski, M., Tait, A. M., Korotkikh, E., 675 Clifford, H., Kang, S., Sherpa, T. C., Singh, P. K., Koch, I., and Birkel, S.: Mt. Everest's highest 676 glacier is a sentinel for accelerating ice loss, npj Clim. Atmos. Sci., 5, 7, 677 https://doi.org/10.1038/s41612-022-00230-0, 2022.

678 Radić, V. and Hock, R.: Regionally differentiated contribution of mountain glaciers and ice caps to future 679 sea-level rise, Nat. Geosci., 4, 91–94, https://doi.org/10.1038/ngeo1052, 2011. 680 Radić, V., Menounos, B., Shea, J., Fitzpatrick, N., Tessema, M. A., and Déry, S. J.: Evaluation of different 681 methods to model near-surface turbulent fluxes for a mountain glacier in the Cariboo Mountains, 682 BC, Canada, The Cryosphere, 11, 2897–2918, https://doi.org/10.5194/tc-11-2897-2017, 2017. 683 Shean, D. E., Bhushan, S., Montesano, P., Rounce, D. R., Arendt, A., and Osmanoglu, B.: A Systematic, 684 Regional Assessment of High Mountain Asia Glacier Mass Balance, Front. Earth Sci., 7, 363, https://doi.org/10.3389/feart.2019.00363, 2020. 685 686 Smeets, C. J. P. P. and van den Broeke, M. R.: The Parameterisation of Scalar Transfer over Rough Ice, 687 Bound.-Lay. Meteorol., 128, 339-355, https://doi.org/10.1007/s10546-008-9292-z, 2008. 688 Suter, S., Hoelzle, M., and Ohmura, A.: Energy balance at a cold Alpine firn saddle, Seserjoch, Monte 689 Rosa, Int. J. Climatol., 24, 1423-1442, https://doi.org/10.1002/joc.1079, 2004. 690 Thibert, E., Dkengne Sielenou, P., Vionnet, V., Eckert, N., and Vincent, C.: Causes of Glacier Melt 691 Extremes the Alps Since 1949, Geophys. Res. Lett.. https://doi.org/10.1002/2017GL076333, 2018. 692 693 van Pelt, W., Pohjola, V., Pettersson, R., Marchenko, S., Kohler, J., Luks, B., Hagen, J. O., Schuler, T. V., 694 Dunse, T., Noël, B., and Reijmer, C.: A long-term dataset of climatic mass balance, snow conditions, 695 and runoff in Svalbard (1957-2018), The Cryosphere, 13, 2259-2280, https://doi.org/10.5194/tc-696 13-2259-2019, 2019. 697 van Pelt, W. J. J., Oerlemans, J., Reijmer, C. H., Pohjola, V. A., Pettersson, R., and van Angelen, J. H.: 698 Simulating melt, runoff and refreezing on Nordenskiöldbreen, Svalbard, using a coupled snow and 699 energy balance model, The Cryosphere, 6, 641–659, <a href="https://doi.org/10.5194/tc-6-641-2012">https://doi.org/10.5194/tc-6-641-2012</a>, 2012. 700 Wang, N., He, J., Pu, J., Jiang, X., and Jing, Z.: Variations in equilibrium line altitude of the Qiyi Glacier, 701 Qilian Mountains, over the past 50 years, Chin. Sci. Bull., 55, 3810-3817, 702 https://doi.org/10.1007/s11434-010-4167-3, 2010. 703 Yang, K., Koike, T., Fujii, H., Tamagawa, K., and Hirose, N.: Improvement of surface flux 704 parametrizations with a turbulence-related length, Q. J. R. Meteorolog. Soc., 128, 2073-2087, 705 https://doi.org/10.1256/003590002320603548, 2002.

706 Yang, W., Guo, X., Yao, T., Yang, K., Zhao, L., Li, S., and Zhu, M.: Summertime surface energy budget 707 and ablation modeling in the ablation zone of a maritime Tibetan glacier, J. Geophys. Res., 116, D14116, https://doi.org/10.1029/2010JD015183, 2011. 708 709 Yao, T., Thompson, L., Yang, W., Yu, W., Gao, Y., Guo, X., Yang, X., Duan, K., Zhao, H., Xu, B., Pu, J., 710 Lu, A., Xiang, Y., Kattel, D. B., and Joswiak, D.: Different glacier status with atmospheric 711 circulations in Tibetan Plateau and surroundings, Nat. Clim. Chang., 2, 663-667, 712 https://doi.org/10.1038/NCLIMATE1580, 2012. 713 Yao, T., Bolch, T., Chen, D., Gao, J., Immerzeel, W., Piao, S., Su, F., Thompson, L., Wada, Y., Wang, L., 714 Wang, T., Wu, G., Xu, B., Yang, W., Zhang, G., and Zhao, P.: The imbalance of the Asian water 715 tower, Nat. Rev. Earth Environ., 3, 618-632, https://doi.org/10.1038/s43017-022-00299-4, 2022. 716 Zhang, G., Kang, S., Fujita, K., Huintjes, E., Xu, J., Yamazaki, T., Haginoya, S., Wei, Y., Scherer, D., 717 Schneider, C., and Yao, T.: Energy and mass balance of Zhadang glacier surface, central Tibetan Plateau, J. Glaciol., 59, 137–148, <a href="https://doi.org/10.3189/2013JoG12J152">https://doi.org/10.3189/2013JoG12J152</a>, 2013. 718 719 Zhang, G., Kang, S., Cuo, L., and Qu, B.: Modeling hydrological process in a glacier basin on the central 720 Tibetan Plateau with a distributed hydrology soil vegetation model, J. Geophys. Res.: Atmos., 121, 721 9521-9539, https://doi.org/10.1002/2016JD025434, 2016. 722 Zhang, T., Deng, G., Liu, X., He, Y., Shen, Q., and Chen, Q.: Heatwave magnitude quantization and 723 impact factors analysis over the Tibetan Plateau, npj Clim. Atmos. Sci., 8, 2, 724 https://doi.org/10.1038/s41612-024-00877-x, 2025. 725 Zhang, Y., Enomoto, H., Ohata, T., Kitabata, H., Kadota, T., and Hirabayashi, Y.: Glacier mass balance 726 and its potential impacts in the Altai Mountains over the period 1990-2011, J. Hydrol., 553, 662-727 677, https://doi.org/10.1016/j.jhydrol.2017.08.026, 2017. 728 Zhao, C., Yang, W., Westoby, M., An, B., Wu, G., Wang, W., Wang, Z., Wang, Y., and Dunning, S.: Brief 729 communication: An approximately 50 Mm3 ice-rock avalanche on 22 March 2021 in the Sedongpu 730 valley, southeastern Tibetan Plateau, The Cryosphere, 16, 1333-1340, https://doi.org/10.5194/tc-731 <u>16-1333-2022</u>, 2022. 732 Zhu, F., Zhu, M., Guo, Y., and Yao, T.: Observation and Simulation of Runoff During an Extreme 733 Heatwave in a Glacial Basin on the Central Tibetan Plateau, Hydrol. Processes, 38, 734 https://doi.org/10.1002/hyp.70014, 2024a.

| 735 | Zhu, F., Zhu, M., Yang, W., Wang, Z., Guo, Y., and Yao, T.: Drivers of the Extreme Early Spring Glacier  |
|-----|----------------------------------------------------------------------------------------------------------|
| 736 | Melt of 2022 on the Central Tibetan Plateau, Earth Space Sci., 11, e2023EA003297,                        |
| 737 | https://doi.org/10.1029/2023EA003297, 2024b.                                                             |
| 738 | Zhu, M., Yao, T., Yang, W., Xu, B., and Wang, X.: Evaluation of Parameterizations of Incoming            |
| 739 | Longwave Radiation in the High-Mountain Region of the Tibetan Plateau, J. Appl. Meteorol.                |
| 740 | Climatol., 56, 833–848, https://doi.org/10.1175/JAMC-D-16-0189.1, 2017.                                  |
| 741 | Zhu, M., Yao, T., Yang, W., Xu, B., Wu, G., and Wang, X.: Differences in mass balance behavior for three |
| 742 | glaciers from different climatic regions on the Tibetan Plateau, Clim. Dyn., 50, 3457-3484,              |
| 743 | https://doi.org/10.1007/s00382-017-3817-4, 2018.                                                         |
| 744 | Zhu, M., Thompson, L. G., Yao, T., Jin, S., Yang, W., Xiang, Y., and Zhao, H.: Opposite mass balance     |
| 745 | variations between glaciers in western Tibet and the western Tien Shan, Global Planet. Change, 220,      |
| 746 | 103997, https://doi.org/10.1016/j.gloplacha.2022.103997, 2023.                                           |
| 747 |                                                                                                          |