# Peer review of "Evaluation of Turbulent Flux Parameterizations over a"

_EGUsphere, 2025_

## Referee Comment (RC2)

**Review of "Evaluation of Turbulent Flux Parameterizations over a Continental Glacier on the Tibetan Plateau"**

This manuscript by Xu et al. provides eddy covariance measurements from Dunde glacier on the Tibetan Plateau, and assesses the performance of five bulk aerodynamic methods to model the measured heat fluxes. The authors explore the seasonal variability of model performances and investigate the simulation of heat fluxes during a heat wave. Among the five methods tested, the authors conclude that the bulk method with a Monin-Obukhov correction performs best. However, there are no details presented on the methods used to process the turbulent flux data, which can significantly alter the resulting measured flux. This fact, along with a handful of methodological ambiguities (e.g., the calculation of roughness lengths), makes the origin of the underwhelming model performance unclear. For these reasons, and more which I detail below, I encourage a reassessment of the methods, analysis, and key outcomes prior to resubmitting.

**General comments**

- 1. As the results of this manuscript hinge on the measured fluxes, it is important to have reliable, reproducible flux measurements. However, there is no mention of how any of the turbulence data are processed. What coordinate system rotation is applied, if any? Is any filtering applied? Are any frequency corrections applied? Over what interval are the fluxes calculated? This last point is particularly important, as there is increasing evidence that 30 min is likely too long an averaging interval to capture turbulent fluxes on glaciers [Mott et al., 2020, Nicholson and Stiperski, 2020, Lord-May and Radić, 2024].
- 2. The methods section lacks a clear flow and connection between the various bulk methods introduced. The connection is not made clear that many of these models differ only in their treatment of stability within the bulk exchange coefficient. I suggest a rewrite with a clear derivation, starting with the  $C_{log}$  method. All assumptions should be clearly presented throughout. You make this connection on L395, but this should be clarified much sooner.
- 3. There are no tests of robustness or statistical significance in this manuscript. Considering how low the average fluxes are, this should be included to instill confidence in the results. At present, I am somewhat wary of the model results, as the models do not appear to accurately simulate the fluxes. For example, the average spring LE is -15.1 W/m², but the best performing model had an RMSE of roughly 8.5 W/m². This is a relative error of over 50%. For H, the relative error of the best-performing model exceeds 100%. Without a more clear breakdown of the flux processing methods, a section discussing modelled roughness lengths, and scatter plots to better present the measured vs. modelled heat fluxes, it's challenging to determine whether this performance is due to the models not applying, or whether there has been a procedural error. I do have several concerns about the application of the bulk methods, which I describe in my specific comments.
- 4. The language needs to be re-evaluated throughout to provide a precise description of the data at hand. For example, words including "marked", "notable", and "pronounced", often overexaggerate throughout this manuscript.
- 5. The structure needs to be revised, as it lacks proper, deep discussion section that explores what these findings mean in a broader context. 5.1 Belongs in the methods, and most of 5.2 and 5.3 are results.
- 6. It is somewhat unclear exactly what knowledge gap this study aims to fill, or how future studies should use their findings. What does it mean for one method to have better spring performance than another? How could a future study leverage this information? This study would benefit from a more systematic exploration of why certain methods outperform others.

**Specific comments**

- L18: Here, and elsewhere: use daily instead of diurnal.
- L53: Estimation using numerical models is not the same as attaining turbulent flux data.
- L62: Is it the processes that aren't well understood? Or the relative contributions of these processes to the overall budget not understood?
- L63: I'm not sure I understand how better flux measurements leads to a better understanding of glacier variations across the TP.
- L67: MO theory is not computationally intensive. Its accuracy, especially on glaciers with strong stability and katabatic winds, has been questioned. The assumptions underpinning MO theory fundamentally break down when fluxes are not constant in height (often the case over glaciers) [Grisogono et al., 2007]. This is part of what the stability corrections aim to address.
- L71: Oerlemans 2000 does not do this.
- L72: Stability functions have existed for many decades to modify turbulent fluxes predicted from bulk methods to account for suppressed fluxes due to stable stratification. Importantly, these functions do not themselves simulate turbulent fluxes. I would suggest Louis 1979 and other fundamental papers here.
- L76: I do not understand this transition how do poorly performing flux models relate to winter and summertime flux measurements?
- L80: I do not agree. Would winter and summertime fluxes not be different because the seasons are different?
- L104: capitalize Northeastern
- L106: Is it gaining mass? I don't understand how one mass balance estimate can come from two separate studies.
- L111: It's more correct to say that fluxes were measured with a sonic anemometer and processed using the eddy covariance method.
- L113: EC is accurate, not particularly precise (see Figure 3).
- L120: Are you assuming the precipitation to be the same between on- and off-glacier sites separated by 350 m vertically? You will need to substantiate this. Further, ERA5 is has been shown to simulate precipitation quite poorly in glaciated environments [e.g., supplemental figure S2 in Draeger et al., 2024]. How did you correct for this? How much data was missing?
- L125/Figure 1: Is the sonic anemometer oriented in the direction of prevailing winds?
- L130/Table 1: Incorrectly identified as table 2. Where are the specifications of the sonic anemometer? Temperature accuracy is incorrect. That is for the operation range above 20 °C. Pascal has units of Pa. Precipitation, what is FS? Why do the heights have different numbers of decimals?
- L133: No overview or context of the methods are provided, instead jumping into naming conventions.
- L156: Constants established how?
- What are your modelled roughness lengths? How do they compare with other studies?
- L157: How do you establish  $\gamma$ ? Is it ever 0?
- L168: How do you calculate Rib?
- L170: The stability functions are typically denoted by  $\phi$ .  $\emptyset$  is the empty set.
- $\bullet\,$  L177:  $\kappa$  is the typical notation for the von Kármán constant.
- L181: The Andreas surface renewal model does not assume these are equal.

- L193: 0.16 does not come out of nowhere here it is  $\kappa^2$ .
- L195: This equation is incorrect. The  $c \sim C^*$  in the Louis model is an empirical constant not  $C_{Hn}$ .
- L200: The ratio of molecular weights is expressed by  $\epsilon$ .  $\in$  denotes set inclusion.
- L211: RMSE, MAD, and MBE are standard measures and don't need to be defined.
- L222: This information was already provided.
- L224: In table 1, you say that pressure is P, not Pres.
- L246: This is fine as a sanity check, but is otherwise a given.
- L255: How do you define the range? 1.5 between min and max? That isn't much, if so.
- L257: What do you mean by stable here?
- L259: How often was the wind aligned with the direction of the glacier slope? Or was the wind more often aligned with the westerlies? Were there katabatic winds? Channelization of flows from above?
- L266: I don't think I would describe variations on this scale as a marked influence. Back-of-the-napkin math suggests density variations of less than 5%.
- L271/Figure 2: How are you calculating the daily statistics? It appears, especially in (k), that the daily averages lead the hourly signals? By visual inspection, it appears that the average WD is calculated directly. WD is a circular variable so averages of cos(WD) and sin(WD) need to be calculated separately. I wonder if demeaned daily cycles would be even more informative?
- L277: This should be explained in your methods.
- L298/Figure 3: How much do you trust these measurements? Is the sensible heat flux really going from  $-150\,\mathrm{Wm}^{-2}$  to  $-20\,\mathrm{Wm}^{-2}$  in the span of an hour? See general comment on flux processing. Figure caption spills onto next page.
- L316/Figure 4: Although time series can be nice to see general trends, they're not informative when making comparisons. I cannot tell which of these models performed best through this figure. A scatter plot would be more helpful. Why are the so few missing measurements here compared to Figure 3?
- L335: Across which meteorological regimes?
- L370: I'm not sure I agree that this is a good alignment. The curves look similar qualitatively, but the relative (%) differences between measured and modelled are substantial.
- L378: Do you mean stable in the sense of atmospheric stability?
- L387: What do you mean by stable here? Furthermore, I do not agree that this is particularly accurate.
- L394: Almost all of this section needs to be reworked, and included in your methods section, see my general point. This is not a discussion of your results.
- L406: What is the relevance of the Greenland Ice Sheet?
- L408:  $C_H$  isn't necessarily a constant. It depends on the roughness length, which you said you weren't fixing on L139.
- L420: The iteration on L in some models is not done to improve responsiveness. L is defined in terms of fluxes, but the fluxes are modelled in terms of L. This circular dependence necessitates an iterative scheme.
- L423: I don't understand why albedo is mentioned here.
- L425: The  $C_{M-O}$  method did not seem to perform best in all cases.
- L427: Most of the schemes used should have different roughness lengths. The C parameters are not constants.

- L429: The iterative algorithm does not ensure energy balance closure.
- L430: How was this ranking done? For example, the  $C_{\rm kat}$  model had lower |MBE|.
- L433: I don't think interseasonal RMSE difference is a very useful statistic. If the RMSE was 100 in all seasons, it would be a terrible model, but the interseasonal RMSE difference would be 0.
- L441: I do not understand what "recalibrated parameters" means in the context of this section. If this relates to EBFM, this needs to be explained in the methods. Because of this, I don't understand how  $T_s$  is being modelled.
- L451/Figure 6: Units missing.
- L455/Table 4: Units missing.
- L457: Are the correlations correct? Comparing the measured to modelled surface height visually, correlations of r > 0.9 is surprising.

**References**

- C. Draeger, V. Radić, R. H. White, and M. A. Tessema. Evaluation of reanalysis data and dynamical downscaling for surface energy balance modeling at mountain glaciers in western canada. *The Cryosphere*, 18(1):17–42, 2024.
- B. Grisogono, L. Kraljević, and A. Jeričević. The low-level katabatic jet height versus monin-obukhov height. Quarterly Journal of the Royal Meteorological Society: A journal of the atmospheric sciences, applied meteorology and physical oceanography, 133(629):2133–2136, 2007.
- C. Lord-May and V. Radić. Improved processing methods for eddy covariance measurements in calculating sensible heat fluxes at glacier surfaces. *Journal of Glaciology*, pages 1–18, 2024.
- R. Mott, I. Stiperski, and L. Nicholson. Spatio-temporal flow variations driving heat exchange processes at a mountain glacier. *The Cryosphere*, 14(12):4699–4718, Dec. 2020. doi: 10.5194/tc-14-4699-2020.
- L. Nicholson and I. Stiperski. Comparison of turbulent structures and energy fluxes over exposed and debris-covered glacier ice. *Journal of Glaciology*, 66(258):543–555, Aug. 2020. doi: 10.1017/jog.2020.23.